# The Impact of Both Individual and Collaborative Job Crafting on Spanish Teachers' Well-Being

**Consuelo Alonso** [1,*], **Samuel Fernández-Salinero** [2] **and Gabriela Topa** [1,*] 

1 Department of Social and Organizational Psychology, The National Distance Education University (UNED), 28040 Madrid, Spain

2 Medicine and Surgery, Psychology, Preventive Medicine and Public Health and Medical Microbiology and Immunology Department, Universidad Rey Juan Carlos, 28933 Madrid, Spain; samuelfssm@gmail.com

* Correspondence: chelo_aa@hotmail.com (C.A.); gtopa@psi.uned.es (G.T.); Tel.: +34-913-988-911 (G.T.)

**Abstract:** Current changes in social structures and political-economic systems directly affect teachers' job performance. Among others, these changes include changes in communication and information technologies, the scientific revolution, changes in the structure of populations, the revolution of social relations, economic and political transformations, and revolutions in labor relations and leisure time. These changes all seem to have promoted educational revolutions, which encourage the development of autonomous individuals who are capable of making critical judgments, ready to dialogue and cooperate in problem solving, and who seek alternatives aimed at building a better society. Thus, teachers suffer daily from the impact of continual changes that affect the way they do their work. According to the job-demands resources model, each job environment has its own characteristics that can be grouped into two dimensions: job demands and job resources. However, the relationship between job demands and resources has serious implications for individuals' lives and psychological well-being. While work provides us with the means to survive, develop social relationships, and experience control over our lives, an excess of demands and a shortage of resources to cope with them would adversely affect personal well-being. Hence, individuals can perform behaviors through job crafting to balance this relationship between demands and resources at work. Job crafting is a proactive behavior of the worker who improves his own working conditions in order to achieve a more meaningful and satisfactory job. This phenomenon allows individuals to play a certain role by "creating" their own job, changing the conditions in which they perform their tasks. In this study, 146 teachers participated to investigate the relationships between both individual and collaborative job crafting behaviors, on the one hand, and job satisfaction, work engagement, and teaching performance, on the other.

**Keywords:** job crafting; work-engagement; job performance; job satisfaction; teachers

## 1. Introduction

Schools and teachers are affected by the transformations that take place in society. Current changes in the social structures and political-economic systems directly affect teachers' job performance. Among others, these changes include changes in communication and information technologies, the scientific revolution, changes in the structure of populations, the revolution of social relations, economic and political transformations, and revolutions in labor relations and leisure time. These changes all seem to have promoted educational revolutions, which encourage the development of autonomous individuals who are capable of making critical judgments, ready to dialogue and cooperate in problem solving, and who seek alternatives aimed at building a better society. Following Di Fabio [1], the current working environment impose complex, intense and contradictory demands on the employees, and this

new scenario affects their professional efficacy as well as their human well-being and development. Moreover, the positive psychology approach [2] encourages research and interventions focused on helping the development of personal resources as well as the construction of healthy working environments at the organizational level.

Thus, teachers suffer the impact of continual changes that affect the way they do their work. Under the positive psychology approach, seminal works have tried to identify the subtle processes that occur between the job features and both positive and negative outcomes, such as stress and engagement [2]. According to the job-demands resources model (JDR; [3]), each job environment has its own characteristics that can be grouped into two dimensions: job demands and job resources [4]. Job demands are the set of physical, psychic, organizational, or social characteristics of a job that require physical and mental effort, which is associated with physiological and psychological costs [5,6]. Job resources refer to the physical, psychic, social, and organizational aspects of work that allow individuals to achieve job goals, reduce job demands and physiological and psychological costs, and lastly, stimulate personal growth, learning, and development [7].

In the research on teachers' professional development within the JDR model [8], two types of job demands have been identified for the collective of teachers. Firstly, there is job pressure, related to quantitative demands, such as the pace and load or volume of the tasks. Secondly, there are the emotional demands resulting from interactions with others. Evers et al. [8] underline the negative impact of job demands on teachers, while stressing certain positive effects. The positive effects of emotional demands consist of stimulating active work because they promote teachers' participation in professional development activities. In relation to the resources of the teaching collective, Evers et al. identify four types of resources: the learning environment, the support and social recognition of an immediate superior, the support and social recognition of more closely related colleagues, and the value of what teachers learn for their teaching job [8].

However, the relationship between job demands and resources has serious implications for individuals' lives and psychological well-being. While work provides us with the means to survive, develop social relationships and experience control over our lives, an excess of demands and a shortage of resources to cope with them would adversely affect personal well-being. Hence, individuals can perform behaviors through job crafting to balance the relationship between demands and resources at work. Job crafting is a set of initiatives of change generated by the person to make the job better fit their expectations and needs [9]. In this sense, Bakker and Demerouti [10] added that it is an individual and proactive action aimed at changing the job to make it more "engaging" and less tiring. This change can be to reduce or modify job demands or to increase the resources available to deal with them. Job crafting is a proactive behavior of the worker who improves his own working conditions [11] in order to achieve a more meaningful and satisfactory job.

Job crafting is individual behavior oriented to changing the limits of one's job through modeling and actively, cognitively, and physically readjusting one's work activity [12]. This phenomenon allows individuals to play a certain role by "creating" their own job, changing the conditions in which they perform their tasks. It is a proactive behavior initiated by the individual [13], not explicit as any job requirement, but which allows people to readjust their competencies, needs, expectations, and job aspirations [12,14]. Job crafting influences what, how, when, and with whom one's job is carried out, modifying its purpose and significance. Thus, job crafting is related to work engagement, job satisfaction, and improvement in the performance of the work itself [15].

Although the scientific literature on job crafting is rapidly increasing, there are still few works developed in Spain [16] and, to date, none has focused on the collective of teachers. Therefore, the final objective of the present work is to analyze the effect of job crafting on job satisfaction, work engagement, and teaching performance. However, as teaching is carried out in the school setting, in which many activities and processes are shared with colleagues, the behaviors of job crafting that the individual may perform are often limited by the collaboration of these colleagues. Therefore, the theoretical model of job crafting, which distinguishes between individual and collaborative

job crafting, seems the most suitable for analyzing teachers' proactive behavior to modify the workplace [12]. The contributions of this study will improve comprehension of teachers' job crafting in Spain, and will allow establishing how individual and collective job crafting behaviors affect teachers' well-being, in both their dimensions of hedonic and eudemonic well-being [11].

## 2. Theoretical Background

More than three decades ago, some authors [16] assumed that workers carry out changes in their jobs on their own initiative, with or without the involvement of their superiors. Thus, workers do not always perform their assigned tasks, but instead they modify their work activity to readjust it to their needs, expectations, and preferences. In particular, job crafting is the proactive behavior in which the individual makes changes in job demands and resources, on his own initiative and voluntarily, to make his work more meaningful, attractive, and satisfactory [17]. For Tims and Bakker [18] job crafting is the set of changes a worker can make to balance job demands and resources with his personal skills and needs.

However, in some jobs, the tasks are not only performed individually, but instead, coworker collaboration is necessary for many of them. For this reason, Leana et al. [12] broadened the concept of job crafting, to include not only the active role of the individual (individual job crafting) but also the behavior by which workers jointly customize and organize their jobs by performing collaborative job crafting. In this second form of job crafting, social links are very relevant. Thus, certain work settings are characterized by social interactions among peers and mutual dependence in task performance. In particular, the frequency and closeness with which these interactions among colleagues develop over time are predictors of job crafting, especially the collaborative type [12].

## 3. Job Crafting and Teachers' Well-Being

Labor organizations are paying increasing attention to the well-being of their workers. The approach of Psychology of Sustainability and Sustainable Development emphasizes the importance of well-being in its hedonic and eudaimonic dimensions [19–21].

Hedonic well-being is understood as self-realization, but also as social integration, such that satisfaction in general and job satisfaction in particular are frequently considered indicators of such well-being. Job satisfaction is the positive perspective that individuals develop of the different elements or characteristics of their workplace and organizational environment [22]. Eudaimonic well-being, on the other hand, includes the behaviors of people who are positively oriented towards other people in their work setting or towards the task and the work organization. Work engagement, or a psychological link with the job, is the positive state of mind associated with the achievement of job goals, and is characterized by vigor, dedication, and absorption [23]. In this sense, work engagement and job performance are often considered indicators of eudaimonic well-being.

Consequently, the literature is accumulating evidence of the impact of job crafting on workers' well-being [24]. In relation to the hedonic dimension of well-being, Wrzesniewski and Dutton [14] defended the relationship between job satisfaction and job crafting. That is, individuals can improve the meaning of their work by modifying their tasks or their interactions. In this way, they could redefine the purpose of their work and their working experience, and this change contributes to generating greater satisfaction.

In relation to the eudemonic dimension of well-being, the evidence points to a positive link between job crafting and work engagement [25,26]. Thus, proactive subjects who perform job crafting behaviors will expand their job resources [27]. Likewise, Xanthopoulou et al. [28] related work engagement with higher success rates in work settings. Person-post fit increased through strategies of job crafting, favoring individuals' well-being and control over the job [29].

Finally, according to various authors, crafting job resources and demands is not only related to work engagement but also to a potential increase in work achievement and performance [30,31]. The research carried out by Leana et al. [12] with teachers and assistants in kindergartens examined

how job crafting affected the quality of their work. Collaborative crafting was positively related to job performance, especially among the more novice teachers (and assistants). It was also associated with high levels of work engagement [11,12,15].

Based on the reviewed literature, the following hypotheses are presented:

**Hypothesis 1.** *Job crafting, both individual and collaborative, will predict job satisfaction.*

**Hypothesis 2.** *Job crafting, both individual and collaborative, will predict work engagement.*

**Hypothesis 3.** *Job crafting, both individual and collaborative, will predict job performance.*

## 4. Methods

### 4.1. Participants

In this study, 146 teachers participated (77.4% females). The sociodemographic characteristics of the sample are shown in Table 1.

**Table 1.** Sociodemographic characteristics of the sample.

| Variables | Categories | Mean | Percentage (%) |
|---|---|---|---|
| Age | | 43.12 | |
| Professional seniority | | 16.22 | |
| Position seniority | | 7.3 | |
| Type of school | Kindergarten | | 11.6 |
| | Primary school | | 38.4 |
| | Professional training | | 2.1 |
| | Secondary school | | 43.8 |
| | University | | 2.1 |
| | Language school | | 0.7 |
| | Musical school | | 0.7 |
| Type of current job | Teaching | | 32.9 |
| | Teaching and tutoring | | 53.4 |
| | Principal and assistants | | 13.0 |
| | Teaching and research | | 0.7 |

### 4.2. Procedure

This study was approved by the committee of ethics of the National Distance Education University in December of 2018. To carry out the research, a questionnaire was developed through the Google Forms platform. As the authors of this research are teachers, they disseminated the invitation to potential participants through their social networks among the teachers of the Andalusian community. The recipients received an email, inviting them to participate, with the explanation of the objectives of the investigation and those that agreed to participate, were requested to send the informed consent before completing the questionnaire. The anonymity of the responses was guaranteed, as the survey did not collect any personal data such as emails the respondents. Data were collected between March and July of 2018. Hence, the final sample is totally intentional. Despite the fact that the sample does not represent the population of Andalusian teachers, and that the work environments of staff in different schools could be quite different, all the organizations belong to the Public Education system in Andalusian community, sharing among them similar legislation, professional background of teachers, and rewards systems. Hence, these similarities allow us to combine them and analyze them together.

## 5. Instruments

Job satisfaction was evaluated with the Brief Affective Job Satisfaction Index in the Spanish version of Fernández and Topa [32], based on Thompson and Phua, [33], made up of four items. Examples of items are: "Most days, I am excited about my work as a teacher", "I feel actually very satisfied with my job", "I like my job most than much of the people", or "I really enjoy the work I do". Responses were collected through a 5-point Likert-type scale ranging from 1 ("totally disagree") to 5 ("totally agree").

Job crafting: We used the scales designed by Leana et al. [12], both with 6 items for each dimension (individual and collaborative job crafting). The items are the same in the two subscales, but the expression "on my own" is used for the individual subscale, and "together" for the collaborative dimension. In the present study, we used the version adapted to Spanish by Llorente and Topa [34], which had adequate psychometric properties. Examples of items are: "You introduce new approaches on your own to improve your work in the classroom", "You change minor job procedures that you think are not productive on your own", "You organize special events in your classroom on your own" (individual job crafting), "Together with your coworkers, you decide to organize special events in your classroom (such as celebrating a child's birthday, etc.)", "You work together with your coworkers to introduce new approaches to improve your work in the classroom", and "You decide together with your coworkers to change the way you do your job to make it easier to yourself" for the collaborative dimension. The questionnaire was rated on a 5-point Likert-type scale ranging from 1 ("never") to 5 ("always").

Work Engagement was evaluated with the Utrecht Work Engagement Scale (UWES 9), [35] in its Spanish adaptation [36]. Examples of items are: "I feel strong and firm in my (teaching) work", "My teaching work inspires me", and "I am immersed in my teaching work." The questionnaire was rated on a 5-point Likert-type scale ranging from 1 ("never") to 5 ("always").

Work performance was assessed using 7 items from the In-Role Behavior scale (IRB) of Williams and Anderson [37]. The original English version, which had already been translated into Spanish in previous studies [38], was adapted to the teaching collective for the present study. Examples of the items are: "I adequately complete the tasks assigned to me in the educational center where I work", "I fulfill the specific responsibilities of my teaching work", or "I am involved in activities that are going to be taken directly into account in the evaluation of my job performance (such as participation in educational, innovation or lifelong learning projects)". The questionnaire was rated on a 5-point Likert-type scale ranging from 1 ("never") to 5 ("always").

*Data Analysis*

Descriptive, correlational analyses and stepwise multiple linear regression analyses were carried out. Age and gender were included as control variables in the first step, individual job crafting and collaborative job crafting in the second step, using the SPSS 25 software (IBM, NewYork, USA).

## 6. Results

First, in the Pearson correlation matrix, positive relationships were revealed between job satisfaction and individual and collaborative job crafting, as well as with work engagement and job performance. Work engagement also presented positive and significant relationships with the indicators of individual and collaborative job crafting. On another hand, work performance presented positive relationships with individual and collaborative job crafting, and with work engagement, as Table 2 displayed.

**Table 2.** Descriptive statistics and Pearson's correlation matrix.

| | Mean | S.D. | 1 | 2 | 3 | 4 | 5 | 6 | 7 |
|---|---|---|---|---|---|---|---|---|---|
| 1. Age (number of years) | 43.12 | 8.5 | - | | | | | | |
| 2. Gender | - | - | 0.058 | - | | | | | |
| 3. Job crafting individual | 3.79 | 0.67 | −0.116 | 0.311 ** | *0.81* | | | | |
| 4. Job crafting collaborative | 3.38 | 0.81 | −0.090 | 0.302 ** | 0.598 ** | *0.88* | | | |
| 5. Job satisfaction | 4.01 | 0.66 | −0.199 * | 0.293 ** | 0.510 ** | 0.456 ** | *0.85* | | |
| 6. Work engagement | 4.14 | 0.71 | −0.279 ** | 0.253 ** | 0.528 ** | 0.402 ** | 0.831 ** | *0.94* | |
| 7. In-role performance | 4.43 | 0.48 | −0.175 * | 0.099 | 0.343 ** | 0.249 ** | 0.349 ** | 0.437 ** | *0.75* |

Note: ** $p < 0.01$, * $p < 0.05$. Values in italics on the diagonal are Cronbach's Alphas.

In relation to the first hypothesis, job crafting explained 31% of the variance of job satisfaction and 33% of work engagement. Lastly, for the model of job performance, a value of $R^2 = 0.11$ was obtained, as Table 3 shows.

**Table 3.** Regression analyses on job satisfaction, work engagement, and in role performance.

| Predictor Variables | Criterion Variables | | | | | |
|---|---|---|---|---|---|---|
| | Job Satisfaction | | Work Engagement | | In-Role Performance | |
| **First Step (Control Variables)** | Beta [a] | | Beta [a] | | Beta [a] | |
| Age | −0.21 ** | −0.15 * | −0.30 ** | −0.22 ** | −0.18 * | −0.14 |
| Gender | 0.30 ** | 0.13 | 0.27 ** | 0.11 | 0.11 | −0.005 |
| **Second step** | | | | | | |
| Job crafting individual | | 0.33 ** | | 00.40 ** | | 0.29 ** |
| Job crafting collaborative | | 0.21 * | | 00.11 | | 0.06 |
| $R^2$ | 0.12 ** | 0.31 ** | 0.14 ** | 0.33 ** | 0.03 * | 0.11 ** |
| $\Delta R^2$ | | 0.20 | | 0.19 | | 0.09 |
| F | 10.74 ** | 20.76 ** | 12.76 ** | 21.18 ** | 3.12 | 7.86 ** |

Note: [a] Standardized regression coefficients, beta. ** $p < 0.01$ * $p < 0.005$.

## 7. Discussion

The findings of this study allow us to confirm the hypotheses. The main goal of this research was to analyze the effect of job crafting, both in its individual and collective dimensions, on teachers' job satisfaction, work engagement, and job performance. We verified the positive influence of job crafting, in both its dimensions, on job satisfaction (Hypothesis 1), secondly, on work engagement (Hypothesis 2), and on job performance (Hypothesis 3).

In the same vein, other investigations have verified that adaptation and customization of job tasks and job conditioners allowed the workers to maintain their well-being, overcome job difficulties, and to grow personally through them [39]. Other works have found that job crafting behaviors can lead to positive results of work engagement, job satisfaction, and resilience at work [40–42]. The present results also coincide with those of previous studies [43], which examined the relationship between job crafting and person-job fit. Thus, if people are in a suitable post, they develop meaningful and valuable work, which favors their emotional well-being and improves their relationships with others.

In the teaching specific setting, the results coincide with the scarce previous studies. Leana et al. [12] explored how teachers developed collaborative job crafting, and under which conditions it was more likely to develop. They also predicted the potential consequences of each condition, confirming the positive relationship of collaborative job crafting with teaching performance, especially for less experienced teachers. These positive results for well-being not only favor the individual but also the organization. In this sense, we underline that these positive results are established in a spiraling cycle along with job demands and resources, as noted by Salanova et al. [44].

This work has several limitations that should be commented on. The most relevant is related to the procedure design and sample selection. As suggested by Podsakoff and colleagues [45], procedural remedies were used to control and counteract common method variance. To reduce evaluation

apprehension and prevent response biases, respondents' anonymity was protected, and they were advised that there were no right or wrong answers. They were requested to respond to the items as honestly as possible. Moreover, we sought to control response consistencies by counterbalancing the item order in the survey, placing the measures of the dependent variables before those of the independent variables [46]. Furthermore, the study variables were measured with validated scales, which can mitigate measurement error and thereby, decrease common method bias [47]. In any case, future research should repeat the study using a longitudinal or time-lagged research design, and gathering data from multiple sources.

The sample selection limits generalization of the conclusions to other groups of teachers. Future research should replicate the results obtained in diverse contexts, including representative samples of teachers from schools of different ownership (public vs. private) and different educational levels. Moreover, our performance assessment has been done by a self-reported measure. Hence, future research should explore more objective measures of performance as days absent, number of complaints among co-workers, among others.

Future research should remedy the above limitations. The present intends to contribute to opening the "black box" of the relationships between situational constraints and personal outcomes, through different processes as both individual and collaborative job crafting. In this sense, both types of proactive behaviors can be considered like new lines of research in the field of organizational psychology. Despite some criticism on the empirical studies on job crafting, blamed as a neoliberal response to the increased job insecurity and loss of meaning at work, the current research on collaborative job crafting highlights the relevance of proactive behaviors in all professional fields. It has been understood as potentially beneficial both for employees and organizations [48], not only in industrial settings but also in the educational [49]. Thus, interventions should be developed to improve teachers' well-being. Among possible measures, workshops to disseminate and apply the behaviors of job crafting could serve as models of action for the entire teaching staff [50].

## 8. Conclusions

The present study investigated the relationships between both individual and collaborative job crafting behaviors, on the one hand, and job satisfaction, work engagement, and teaching performance, on the other. The results have found enough empirical support for the proposed hypotheses.

**Author Contributions:** Conceptualization, C.A. and G.T.; methodology, C.A. and G.T.; software, C.A.; validation, C.A., S.F-S. and G.T.; formal analysis, C.A. and G.T.; investigation, C.A.; resources, G.T.; data curation, C.A.; writing—original draft preparation, C.A.; writing—review and editing, C.A., S.F-S. and G.T.; visualization, C.A.; supervision, G.T.

**Funding:** This research received no external funding.

**Conflicts of Interest:** The authors declare no conflict of interest.

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
