# Peer review of "The Impact of Both Individual and Collaborative Job Crafting on Spanish Teachers’ Well-Being"

_education, doi:10.3390/educsci9020074_

Round 1

Reviewer 1 Report

The theoretical model and the finalities of the research are especially current and interesting, but the methodological section needs to be improved.

Some suggestions:

Line 41: It is possible to add some information and references on this model? Why has it been chosen?Which field of study and research does it to refer to?

Line 63: Which are the theoretical bases or evidence to support such statement?
Line 72: Briefly, explain the connections outlined.

An integration of the methodological section ("Instruments" paragraph) is advisable, adding some items from the scales used. This could be useful to better understand how such complex concepts (job satisfaction; job crafting...) can be analyzed.

Finally, given the great impact which job crafting has on theachers' jobs, it might be useful to examin in depth the characteristics (differences and similarities) of individual and collaborative job crafting.

Author Response

Dear reviewer, we have tried to address your concerns. Many, thanks. 

Reviewer 2 Report

The rationale of doing the current research can be strengthened.

The development of the hypotheses can be improved.

For data collection, the work environments of staff in different type of schools could be quite different. If the authors want to combine them together, a justification may be needed.

The performance is measured with self report. Will this be an issue in studying the relationship between the variables.

The response of (never) to (always) may not match with some of the questions. For instance, "Most days, I am excited about my work as a teacher."

For data analysis, stepwise or hierarchical regression is used? Some of the results reported and the numbers in the table do not match.  

In table 2, "," should be "."  

Author Response

(The authors gave the same response as above.)

Reviewer 3 Report

The research is theoretically well justified and makes sense in the theoretical context of occupational health psychology and sustainability psychology. In addition, the results collected are of great theoretical and practical importance, since they reinforce the model of demand-resources and point to possible ways to achieve the well-being and happiness of teachers.

However, I believe that the work requires some improvements, fundamentally two: (1) To specify that what they measure is subjective performance, and (2) to make explicit in the procedure the mode of selection of the sample (subject this that I comment more extensively next).

The purpose of any statistical analysis is to describe, on the basis of the characteristics observed in the sample, the population to which it belongs. The inference capacity of the research results is very much determined by the size of the sample, in addition to the procedure followed in its selection: whether it has been random or not. Randomness guarantees equality in the probability of being chosen to participate in the sample, of all the units of the population of interest. In this case, we do not know the procedure followed to select the sample, although due to its size and what the authors expressed (line 211 onwards) we can assume that it was intentional, which seriously limits the inferences that can be made from the results. The size of the sample is not the only problem that affects the inference capacity of the results, also the composition of the sample is a problem; e.g., the cases of university professors represent only 2.1% of the sample, in addition, the proportion of women in the sample is very unequal with respect to the proportion of women in secondary and university education in Spain. I believe that it is not enough to recognize these limitations, but that it is also necessary to express to which population and circumstances the results obtained are extensible.

Author Response

(The authors gave the same response as above.)
